# Anti-Inflammatory Activities of *Euglena gracilis* Extracts

**DOI:** 10.3390/microorganisms9102058

**Published:** 2021-09-29

**Authors:** Paola Brun, Anna Piovan, Rosy Caniato, Vanessa Dalla Costa, Anthony Pauletto, Raffaella Filippini

**Affiliations:** 1Department of Molecular Medicine, Section of Microbiology, University of Padova, Via A. Gabelli 63, 35127 Padova, Italy; anthony.pauletto@studenti.unipd.it; 2Department of Pharmaceutical and Pharmacological Sciences, University of Padova, Via F. Marzolo 5, 35131 Padova, Italy; anna.piovan@unipd.it (A.P.); rosy.caniato@unipd.it (R.C.); vanessa.dallacosta@studenti.unipd.it (V.D.C.); raffaella.filippini@unipd.it (R.F.)

**Keywords:** microalgae, lipopolysaccharide, inflammation, reactive oxygen species, novel food

## Abstract

Dietary supplementation with nutrients able to control intestinal and systemic inflammation is of marketable interest. Indeed, gastrointestinal homeostasis plays a significant role in maintaining human health. In this setting, *E. gracilis* may sustain or promote human health, but the effects on the intestinal inflammatory milieu are not clear. In this study, we investigated the anti-inflammatory activity of *E. gracilis* and inferred possible mechanisms. Paramylon, crude, and fractionated extracts were obtained from *E. gracilis* grown in vitro. Phytoconstituents of the extracts were characterized using TLC and HPLC UV-Vis. The anti-inflammatory and antioxidant activities were investigated in primary human macrophages and an intestinal epithelial cell line (HT-29). The analysis of the extracts led to identifying β-carotene, neoxanthin, diadinoxanthin, canthaxanthin, and breakdown products such as pheophytins and pheophorbides. *E. gracilis* fractionated extracts reduced the production of tumor necrosis factor-α triggered by bacterial lipopolysaccharide (LPS) in the short and long terms. Pheophytin a and b and canthaxanthin increased the intracellular reducing potential and dampened the production of LPS-induced reactive oxygen species and lipid peroxidation, intracellular events usually involved in the perpetuation of chronic inflammatory disorders. This study rationalizes the role of specific extract fractions of *E. gracilis* in controlling LPS-driven intestinal inflammation.

## 1. Introduction

Microalgae are industrially valuable microorganisms with great metabolic potential. Indeed, microscopic algae biosynthesize, metabolize, accumulate, and secrete proteins, fatty acids, polysaccharides, carotenoids, phycobilins, vitamins, and sterols more efficiently than traditional crops. In recent years, primary and secondary metabolites from microalgae have been used in the food, pharmaceutical, and cosmetic industries [1], where microalgae have competitive advantages over other plant sources since they grow in soilless medium without seasonal limitations. In this context, the European Union encourages the Bioeconomy, a strategy where algae represent promising biological resources with potential applications in different fields [2]. Indeed, microalgae-based high-value molecules (such as xanthophylls and β-carotene) have considerable market potential, and the development of novel foods based on microalgal biomass is an exciting tool providing nutritional supplements with biologically active compounds. Therefore, the industrial cultivation of microalgae has increased dramatically over the last few decades [3].

*Euglena gracilis* Klebs (Euglenozoa, Euglenophyceae) is a freshwater flagellated unicellular microalga representing one of the simplest and earliest derived eukaryotic cells. It grows under heterotrophic, photoautotrophic, or photoheterotrophic conditions in anaerobically or aerobically mode [1]. Moreover, like most other microalgae, *E. gracilis* contains several bioactive compounds that make it a new functional food, meeting the requirements for tackling malnutrition in developing countries and sustaining human health. Indeed, on 4 May 2020, the EFSA Panel on Nutrition, Novel Foods and Food Allergens (NDA) approved the *Euglena gracilis* dried whole cells as a novel food [4].

In addition to fatty acids, carotenoids, and proteins [5], *E. gracilis* is a good source of paramylon, a linear β-(1, 3)glucan polysaccharide with high molecular weight [6,7] considered a functional dietary fiber for health purposes. Numerous studies have evaluated the effects of *E. gracilis* on human health, but only a few reports have addressed the biological activity of paramylon or specific *E. gracilis* extracts [8,9,10,11,12,13,14,15]. Oral administration of *E. gracilis* improved the glycemic control in a rat model of type 2 diabetes mellitus, whereas the aqueous extract from the microalga inhibited lipid accumulation in cultured human adipose-derived stem cells [14,16]. According to recent experimental studies, paramylon modulates the immune system response by reducing the production of pro-inflammatory cytokines and disease evidence in an animal model of arthritis [17] and atopic dermatitis [18]. Moreover, paramylon was supposed to inhibit gene expression of peroxisome proliferator-activated receptor-γ, a transcription factor emerging as the checkpoint for setting the inflammatory milieu during obesity [16,19].

Because of the anti-inflammatory effect, supplementation with *E. gracilis* is a promising approach for treating overweight and obesity—metabolic disorders characterized by increasing prevalence in the youth population [20]. Indeed, it is well accepted that low-grade systemic inflammation is strictly associated with the onset of obesity and obesity-related diseases such as atherosclerosis, cardiovascular risk, hepatic steatosis, and diabetes [21]. During metabolic disorders, inflammation disrupts intestinal tight junctions and leads to a leaky gut barrier, allowing increased circulating levels of intestinal bacteria-derived lipopolysaccharide (LPS) [22]. The resulting metabolic endotoxemia amplifies inflammation at local and systemic compartments, and it recruits activated macrophages in the gut wall, visceral fat, liver, and joints [23]. However, the increased circulating levels of LPS are not such a rare event, even in lean and healthy subjects. Chylomicrons are preferential shuttles of LPS, and the occasional intake of fatty meals allows the passage of intestinal LPS into circulation. Impaired lipoprotein metabolism reduces LPS catabolism and increases time-limited endotoxemia episodes, eventually leading to inflammatory-related disorders [23,24]. During inflammation, cells generate high levels of reactive oxygen species (ROS) central to endothelial dysfunction and tissue injury, thus accelerating the progression of obesity, atherosclerosis, and type 2 diabetes [25]. However, ROS are normal metabolic byproducts involved in modulating cellular functions such as, paradoxically, signaling pathways enhancing insulin sensitivity in response to physiological stimuli [26]. In eukaryotic cells usually endowed with antioxidant mechanisms and free radical scavengers, the reducing potential sets the hormetic phenomenon of the ROS, defined by beneficial effects at low doses and detrimental effects at higher concentrations.

Considering the requirement of novel functional food characterized by high metabolic potential, in this study we investigated the anti-inflammatory effects of paramylon and crude and fractionated extracts of *E. gracilis*. We did not collect evidence for the anti-inflammatory effects of paramylon. On the contrary, we revealed that in cultured human primary macrophages and intestinal epithelial cell line, specific fractionated extracts of *E. gracilis* increased the cellular threshold for ROS activation under normal conditions, leading to reduced oxidative burst and pro-inflammatory activation triggered by LPS.

## 2. Materials and Methods

### 2.1. Cultures of Euglena gracilis

*E. gracilis* strain (1224-5/27) was obtained from the Culture Collection of Algae (SAG; Göttingen, Germany). Cells were cultured in CM liquid medium [27] supplemented with sodium acetate 3 g/L, in non-agitated Erlenmeyer flasks. The pH was adjusted to 6.8. The cultures were maintained at 24–26 °C under a photoperiod (16/8 h light–dark cycle). A growth curve was calculated by determining the fresh weight every second or third day over a cultivation period of 14 days. The experiment was repeated three times. As soon as the cultures reached the stationary growth phase, an inoculum was transferred in a fresh medium, whereas the remaining algal cells were harvested by centrifugation and concentrated to a biomass slurry that was washed twice with deionized water. Finally, the cleaned samples were stored at −20 °C until use.

### 2.2. Preparation of Paramylon and Crude and Fractionated Extracts from Euglena gracilis Cells

To obtain paramylon (P), cells were extracted three times with acetone in an ultrasound bath for 20 min at room temperature. After centrifugation, the pellet was washed repeatedly with deionized water containing 10% sodium dodecyl sulfate in a water bath at 100 °C. The solid residue, paramylon, was washed with ethanol to eliminate water and was freeze-dried [28]. To obtain the crude extract (CE), cells were extracted three times with acetone in an ultrasound bath for 20 min at room temperature. After centrifugation, the combined acetone extracts were evaporated to dryness. The fractionated extract (FE) was obtained from CE—an aliquot of CE was suspended in methanol/water (40:60) and extracted by ethyl acetate. The ethyl acetate phase was separated and evaporated to dryness. FE was fractionated by analytical TLC, using silica gel 60 F254 pre-coated plates (Merck, Milan, Italy) and acetone/n-hexane (4:6) as eluent to obtain sub-fractionated extracts (sFE1-8).

The analysis of the extracts was performed using an Agilent 1100 HPLC Series System (Agilent, Santa Clara, CA, USA) equipped with degasser, quaternary gradient pump, column thermostat, and UV-Vis detector. A Gemini 5 µm C6-Phenyl column (250 × 4.6 mm) from Phenomenex (Torrance, CA, USA) was set at 40 °C. Analyses were conducted in the isocratic mode, using acetonitrile/methanol (10:90; *v/v*) at a flow rate of 1 mL min^−1^, with an injection volume of 10 µL; detection was performed at 280, 365, 438, and 460 nm. Spectra were acquired from 200 to 800 nm. The identification of the compounds was performed by comparing TLC and HPLC chromatograms and UV-Vis spectra using the available reference standards (zeaxanthin, canthaxanthin, β-carotene: Sigma-Aldrich, Milan, Italy) and data obtained from published references [29,30,31,32,33,34]. Chlorophyll a and chlorophyll b content was determined as described by Yang et al. [30]. Pheophytin content was measured according to the method of Lichtenthaler [29], as reported by Yang et al. [30]. One aliquot of the extract was solubilized with acetone/water (4:1) and, after appropriate dilution, the maximum absorbance was read at 663 nm, 646 nm, 665 nm, and 653 nm for chlorophyll a, chlorophyll b, pheophytin a, and pheophytin b, respectively. The content of pigments was calculated using the following equations:Chlorophyll a (µg/mL) = 12.25 A663 − 2.55 A646
Chlorophyll b (µg/mL) = 20.31 A646 − 4.91 A663
Pheophytin a (μg/mL) = 22.42 A665 − 6.81 A653
Pheophytin b (μg/mL) = 40.17 A653 − 18.58 A665

Since standards were not available for all carotenoids, zeaxanthin and β-carotene were selected as external standards for the quantification. Stock solutions (1 mg/mL) were prepared in acetonitrile/methanol (10:90; *v/v*), and the calibration curves were obtained in a concentration range, respectively, of 0.25–25 μg/mL (R^2^ = 0.9999), 1–100 μg/mL (R^2^ = 0.9996), with six concentration levels. Calibration curves were constructed by plotting the peak area at 438 nm vs. the pigment concentrations. Xanthophylls were quantified as zeaxanthin equivalents. The amount of the compounds was expressed as mg/g of the dry weight of the extract.

For in vitro biological experiments, CE and FE were dissolved in DMSO at 1 mg/mL; P was dissolved at 1 mg/mL in 50 mM Tris HCl pH 7.5, 1 M sucrose, 25 mM NaF, and 3 mg/mL CHAPS prepared in pyrogen-free water. Endotoxins levels were assessed in all extracts using LAL Chromogenic Endotoxin Quantitation Kit (Pierce; Thermo Fisher Scientific, Monza, Italy) and following the vendor instructions. Working solutions were prepared in cell culture media.

### 2.3. Human Cell Culture Conditions

Primary human monocytes were isolated from buffy-coat preparations of whole blood obtained from the blood bank of the Padova University Hospital. The study protocol was reviewed and approved by the hospital’s ethics committee of the Padova University Hospital (registration number CE: 091/2016). The buffy-coat was mixed 1:1 (vol/vol) with sterile RPMI 1640 Medium (Gibco; Life Technologies, Monza, Italy) and layered over Ficoll-Paque PLUS (GE Healthcare; Sigma, Milan, Italy). Samples were centrifuged (1200 rpm, 30 min), and peripheral blood mononuclear cells (PBMCs) were collected. Cells were washed (1600 rpm, 10 min), suspended in RPMI 1640, supplemented with 1% penicillin–streptomycin and 10% heat-inactivated fetal bovine serum (FBS; Life Technologies), and counted. PBMCs were seeded (3 × 10^5^ cells/well) in 96-well tissue-culture plates (Corning; Sigma; Milan; Italy) and cultured for 3 h in a 5% CO_2_, humidified, 37 °C incubator. Floating cells were then removed, whereas attached cells were washed and differentiated in mature macrophages by incubation for 10 days with complete medium supplemented with recombinant human granulocyte–macrophage colony-stimulating factor (rhGM-CSF, 2 ng/mL; ImmunoTools; Friesoythe, Germany). The culture medium was renewed every 3 days.

HT-29 human intestinal epithelial cell line was purchased from ATCC (LGC Standards; Milan, Italy) and cultured in Dulbecco’s modified Eagle medium (DMEM) containing 1% penicillin–streptomycin and 10% of FBS (all provided by Life Technologies). HT-29 were seeded at 2 × 10^4^ cells/well in 96-well tissue culture plates or at 2 × 10^5^ cells/well in 24-well tissue-culture plates (Corning).

### 2.4. Cell Viability Assay

Differentiated primary macrophages or HT-29 were incubated for 24 h with *E. gracilis* extracts at final concentrations ranging from 0 to 100 µg/mL. Control cells were incubated with respective vehicles (see Section 2.2) at the highest final volumes used. At the end of incubation, MTT solution (50 µg/100 µL; Sigma) was added, and cells were incubated for 4 h at 37 °C. Formazan crystals were then solubilized in 100 µL of SDS 10% w/vol, HCl 0.01 N. The absorbance was recorded 16 h later at 590 nm using a microplate reader (Sunrise; Tecan, Männedorf (Zürich); Switzerland). IC_50_ values were determined in three separate experiments, each one performed in duplicate.

### 2.5. Enzyme-Linked Immunosorbent Assay

Differentiated primary human macrophages were incubated for 24 h with *E. gracilis* extracts at final concentrations ranging 0–10 µg/mL with or without 100 ng/mL lipopolysaccharide (LPS from *Salmonella enterica* serotype Typhimurium; Sigma). In different experiments, macrophages were pretreated for 3 days with 0–10 µg/mL of *E. gracilis* extracts. As a positive control, cells were incubated with dexamethasone (1 μg/mL; Sigma). Stimuli were renewed every day, and cells were finally challenged with LPS for 24 h. At the end of incubation, conditioned media were collected and stored at −80 °C. Levels of tumor necrosis factor (TNF)-α were measured in the conditioned media using commercially available enzyme-linked immunosorbent assay kits (ELISA, Affymetrix eBioscience; Prodotti Gianni, Milan, Italy) and developed using 3,3′,5,5′-tetramethylbenzidine (TMB). Optical densities were measured at 450 nm using a microplate reader (Sunrise). The sensitivity of the assay was 15 pg/mL. Experiments were performed in triplicate.

### 2.6. Assessment of Intracellular Glutathione Content

Intracellular glutathione (GSH) levels were assessed in HT-29 cells using Ellman’s method [35]. Cells were cultures in 24-well plates and treated for 4 days with 10 µg/mL of extracts obtained from *E. gracilis*. Stimuli were renewed every day. Cells were collected and incubated with 20% w/vol cold trichloroacetic acid for 30 min. Samples were centrifuged (13,000 rpm for 5 min) and added with 10% vol/vol 5,5′-dithiobis-(2-nitrobenzoic acid) (DTNB). The absorbance was recorded at 412 nm. Values were plotted on a standard curve obtained by serial dilution of *N*-acetylcysteine (Sigma).

### 2.7. Detection of Intracellular Reactive Oxygen Species (ROS) Levels

HT-29 seeded in 24-well plates were pretreated with 10 µg/mL of extracts from *E. gracilis* for 3 days. Stimuli were renewed every day. When described, cells were incubated with LPS 100 ng/mL for 24 h. Cells were loaded for 30 min at 37 °C with 10 µM 2′,7′-dichlorodihydrofluorescein diacetate (H_2_DCFDA; Molecular Probes, Invitrogen, Italy) in warm PBS [36]. Cells were washed twice, harvested by Trypsin-EDTA, washed, and analyzed using BD FACSCalibur flow cytometer (Becton Dickinson, Franklin Lakes, NJ, USA). Ten thousand events were acquired for each experimental condition. Results were analyzed using the WinMDI 2.9 program (Windows Multiple Document Interface for Flow Cytometry).

### 2.8. Evaluation of Lipid Peroxidation

Lipid peroxidation was determined by measuring the thiobarbituric acid (TBA, Inalco, Italy) reactive substances (TBA reactants). HT-29 seeded in 24-well plates was pretreated with 10 µg/mL of *E. gracilis* extracts and then challenged with 100 ng/mL LPS, as described above. Cells were then added with 20% w/vol cold trichloroacetic acid and centrifuged. Supernatants were incubated with 0.67% solution of TBA at 100 °C for 10 min. The absorbance was recorded at 532 nm. Values were plotted on a standard curve obtained by serial dilution of malonaldehyde tetrabutylammonium salt (Sigma). TBA reactant levels were calculated using a molar extinction coefficient of 1.56 × 10^5^ M^−1^ × cm^−1^ [36]. TBA reactant levels were normalized to protein concentration determined by bicinchoninic acid assay (Pierce; Thermo Fisher Scientific, Milan, Italy).

### 2.9. Statistical Analysis

Results are reported as mean ± SE. Graphs were generated using GraphPad Prism 3.03 (San Diego, CA, USA). Statistical significance was calculated using one-way ANOVA test followed by the Newman–Keuls post hoc test. *p* < 0.05 was considered statistically significant.

## 3. Results

### 3.1. Analysis of Euglena gracilis Extracts and Biological Activities

Depending on the solvents used and the protocols, the chemical profile and the biological activities of the extracts can vary greatly. Indeed, the fractionation process can produce more or less active fractions, and the purification process could remove inactive extracts, bringing out the active components. To investigate the most convenient extracting protocol for *E. gracilis*, in this study we evaluated three methods that were applied on *E. gracilis* cultured cells to yield the paramylon (P), the crude extract (CE), and the fractionated extract (FE). The CE fraction analysis led to identifying neoxanthin (1), diadinoxanthin (2), zeaxanthin (3), canthaxanthin (4), chlorophyll b (5), chlorophyll a (6), and β-carotene (7); unidentified xanthophylls were also detected. β-carotene and the same xanthophylls retrieved in CE were identified in FE. In addition, the analysis of FE reported chlorophyll breakdown products such as pheophytin b (8), pheophytin a (9), and pheophorbides (Table 1, Figure 1A,C).

Chlorophyll a, chlorophyll b, pheophytin a, pheophytin b, and total carotenoids were determined in CE and FE fractions (Figure 1). The identified compounds are expressed as mg/g of dry extract and the relative content of the carotenoids in CE and FE (Figure 1B,D). As reported in Figure 1B,D, diadinoxanthin represented about the 56% of total carotenoids in CE and FE fractions; β carotene was also recovered in both fractions. CE fraction differed from the FE fraction for the higher content in neoxanthin.

Extracts P, CE, and FE were assessed for anti-inflammatory activity. As determined by ELISA, cultured human primary macrophages stimulated with LPS significantly increased the production of TNF-α (648.9 ± 46.87 pg/mL) as compared with unstimulated macrophages (115.1 ± 53.27 pg/mL; *p* < 0.01). The levels of TNF-α induced by LPS were significantly reduced only in macrophages incubated with FE, 10 µg/mL (Figure 2A). We did not observe any changes in TNF-α production in macrophages incubated with P and CE extracts. As reported in the literature, LPS triggers reactive oxygen species (ROS) involved in intracellular signals that eventually result in inflammatory cytokine production [37]. We, therefore, investigated whether extracts P, CE, and FE reduced the intracellular ROS levels ignited by LPS stimulation. As reported in Figure 2B, LPS significantly increased ROS levels in HT-29 cells loaded with H_2_DCFDA, a fluorescent probe detecting intracellular ROS. FE significantly reduced intracellular ROS in cells stimulated with LPS, whereas P and CE were not effective.

Considering these results, FE was fractionated by TLC, obtaining the sub-fractionated extracts (sFE) 1 ÷ 8. Figure 3 reports the chemical constituents of sFE. Chromatograms of sFE1 ÷ sFE8 are reported in Appendix A.

### 3.2. Identification of Non-Cytotoxic Concentrations of Extracts from E. gracilis

The cytotoxic activity of extracts obtained from *E. gracilis* was assessed on cultured human primary macrophages and HT-29 cells by the MTT assay. Extracts were tested at concentrations ranging from 0 to 100 µg/mL, and the calculated IC_50_ values are listed in Table 2. P, CE, and FE reported IC_50_ values higher than 30 µg/mL. Sub-fractionated extracts reported no reduction in cell viability at the highest concentration tested (100 µg/mL; Table 2). No cytotoxic activity was detected in cells incubated with the vehicles (data not shown). Considering these results, we decided to perform the subsequent experiments using extracts at concentrations 10 µg/mL or lower.

### 3.3. Sub-Fractionated Extracts from E. gracilis Dampen LPS-Induced Inflammation

We determined by LAL test the endotoxin levels in *E. gracilis* extracts, sub-fractionated extracts, and vehicles, and we found that the levels of endotoxins were below 0.1 EU/mL, corresponding to 0.01 ng/mL (data not shown). As reported by extracts, sub-fractionated extracts obtained from FE were tested for the anti-inflammatory activity in cultured human primary macrophages stimulated with LPS. As reported in Figure 4, all the sFE reduced the production of TNF-α but sFE6 and sFE7. For the sFE endowed with anti-inflammatory activity, the extent of TNF-α reduction was comparable to FE.

In dose–response experiments, we revealed that different extracts from *E. gracilis* preserved the anti-inflammatory activity at concentrations lower than 10 µg/mL. Results of dose–response experiments are reported in Table 3 as the lowest concentration of extracts that can reduce by 25% the production of TNF-α triggered by LPS. As noted, FE, sFE1, sFE2, sFE4, sFE5, and sFE8 significantly reduced production of TNF-α at concentrations lower than 10 µg/mL (*p* < 0.05). sFE1 was the most effective, as the anti-inflammatory activity was evident at 0.1 µg/mL (Table 3, 24 h).

Since *E. gracilis* might be proposed as a functional food for long-term consumption to prevent gut-derived systemic inflammation, in a parallel set of experiments, we investigated whether *E. gracilis* extracts maintain the anti-inflammatory activity at a longer time of exposure, with no significant alteration in the extent of the anti-inflammatory effect. Therefore, human primary macrophages were pretreated for 3 days with FE or sFE and finally challenged with LPS for 24 h. In human macrophages pretreated with *E. gracilis*, LPS increased production of TNF-α (562.2 ± 89.31 pg/mL) at a similar extent observed in not pretreated cells (data not shown), ruling out the hypothesis of functional unresponsiveness or down stimulation caused in macrophages by continuous exposure to *E. gracilis* extracts. Moreover, long-term exposure to *E. gracilis* did not significantly alter the concentration of extracts able to reduce by 25% the production of LPS-induced TNF-α (Table 2, 4 days). No significant increase in TNF-α production was observed in human macrophages incubated for 24 h or 4 days with FE or sFE without LPS (data not shown).

### 3.4. Extracts from E. gracilis Increase the Intracellular Reducing Potential

Increased GSH levels ensure the antioxidant potential in cells, providing additional mechanisms for ROS scavengers. In our experiments, HT-29 cells were exposed to FE and sFE endowed with anti-inflammatory activity (Figure 4). Following 4 days of stimuli, cells reported increased intracellular GSH. In particular, FE, sFE2, sFE3, and sFE5 were the most effective in improving the reducing potential in the HT-29 cell line (Figure 5).

### 3.5. Extracts from E. gracilis Reduce Intracellular Reactive Oxygen Species Generated by LPS

As reported in Figure 6, FE significantly reduced intracellular ROS in HT-29 cells stimulated with LPS. At the same, sFE2, sFE3, and sFE5, the sub-fractionated extracts reported to reduce TNF-α production (Figure 4), dampened the intracellular reactive oxygen burst generated by LPS (Figure 6B). In accordance with data on GSH intracellular content (Figure 5), sFE1, sFE4, and sFE8 did not report antioxidative capability in LPS-stimulated cells (Figure 6A). Among the sFE, sFE3 significantly differed from FE (*p* < 0.05), suggesting that this extract’s antioxidative capability was reduced compared with FE, even if still effective. Incubation of HT-29 cells with extracts alone (10 µg/mL) did not affect intracellular ROS generation (data not shown).

### 3.6. Extracts from E. gracilis Prevent Lipid Peroxidation

In addition to producing pro-inflammatory mediators, the generation of intracellular ROS is usually involved in the degradation of protein, oxidation of DNA, and lipid peroxidation, events frequently involved in the amplification of inflammation and the onset of chronic disorders [28]. Therefore, we evaluated the ability of FE and sFE to reduce ROS production (Figure 6) and prevent lipid peroxidation in LPS-activated HT-29 cells. TBA-reactive substances, a marker of lipid peroxidation, increased in HT-29 cells incubated with LPS (Figure 7). In contrast, lipid damage was blunted in cells pretreated for 3 days with FE, sFE2, and sFE5, indicating a protective effect of specific extracts from *E. gracilis* in LPS-induced oxidative burst.

## 4. Discussion

Several experimental studies have so far evaluated the biological activity of *E. gracilis*, tested as a whole dried powder, with paramylon, water, ethanol, or dichloromethane extracts reporting antitumoral, anti-inflammatory, antioxidative, or proliferative effects [11,12,13,15,17,18]. However, fractions or whole preparations significantly differ in biological activities, as the composition of the extracts depends on the protocol used, the quality of the original material, and any prior treatment [8]. In this study, we investigated the activity of paramylon, acetone crude, and fractionated extracts from cultured cells of *E. gracilis*, a condition that led us to rule out any unnecessary pretreatment of the biological material. The anti-inflammatory and antioxidant activities on primary macrophages and intestinal epithelial cell line were determined for each fraction to identify extracts endowed with increased or reduced activities compared with the crude extract.

In our protocol, we used acetone as the solvent to obtain the crude extract. Indeed, we recently reported that acetone extract of Euglena is endowed with biological activity and modulated microglial activation by inhibiting the expression and release of pro-inflammatory molecules in LPS-induced neuroinflammation [38]. Moreover, acetone extracts photosynthetic pigments with a wide range of polarity and yields fractions enriched in carotenoids [39]. Finally, acetone precipitates paramylon [39], giving us the possibility to test the antioxidant and anti-inflammatory activities of *E. gracilis* in paramylon-free fractions [29,30,31,32,33,34].

Whereas paramylon has been indicated as the most important antioxidant fraction in *E. gracilis*, in this study we were unable to associate any anti-inflammatory and antioxidant activities with the paramylon fraction (Figure 2). In paramylon-free fractions, subsequent extractions led us to identify sFE1 as being most enriched in β-carotene. Carotene plays a variety of essential functions, and the multiple benefits of carotenoids for human health are supposedly linked to their antioxidant and anti-inflammatory effects [40]. The sFE1 fraction reported anti-inflammatory but not antioxidant activities (Figure 4 and Figure 6), confirming a biological response to ROS-independent inflammatory stimuli [41,42]. Our study identified specific sFE from *E. gracilis* enriched in pheophytin a (sFE2), canthaxanthin (sFE5), and pheophorbides (sFE8) endowed with significant anti-inflammatory activity, as determined by the reduction in TNF-α production in LPS-activated human primary macrophages (Figure 4). Some of the sFE increased the reducing potential in cells, demonstrating antioxidative effects to counteract LPS-induced ROS levels and cell damage (Figure 5, Figure 6 and Figure 7).

The subfractionation process of the FE into sFE2, sFE3, and sFE8 resulted in the breakdown of chlorophylls into pheophytins and pheophorbides. Interestingly, the digestion of chlorophylls in the human intestine yields pheophytin and pheophorbide too [43], suggesting the gut as the ideal bioreactor able to increase the biological effects of *E. gracilis* naturally. Pheophytins and pheophorbides are anti-inflammatory agents able to suppress inflammation via reducing nitric oxide production [44]. Here we report that pheophytins also quenched reactive oxygen species (Figure 6) and increased the reducing potential in cells (Figure 5). However, pheophytin a (sFE2) only prevents LPS-induced lipid peroxidation (Figure 7), revealing a more meaningful activity for pheophytin a or different stability [45]. The pheophorbides in sFE8 reported a substantial reduction in TNF-α production, with no antioxidative potential. Indeed, TNF-α is involved in cancer promotion and progression, and pheophorbides have shown important antitumor activities [43,46].

Our results integrate a growing body of knowledge about the use of *E. gracilis* as a functional food. The biotechnological applications of *E. gracilis* allow the production of different metabolites that, together with antioxidants, pigments, minerals, and vitamins, compose a mixture that has been poorly characterized for its chemical composition and biological effects. The assessment of the component–function relationship and mechanisms of action has therefore generated confounding results and controversies. Paramylon, the unique component of *E. gracilis*, is not an exception. Oral administration of this linear β-(1,3)-glucan polymer has been reported to prevent acute hepatic injuries induced by CCl_4_ administration via antioxidative mechanisms and to inhibit atopic dermatitis in mice to a similar or greater extent than prednisolone [11,18]. In vitro experiments have reported that paramylon activates pro-inflammatory pathways and ROS production in neutrophils, monocytes, macrophages, and lymphocytes [15,47]. Moreover, several different compounds have been reported in extracts from *Euglena*, depending on the protocols and techniques [48,49]. The opposite results can be explained by different biological experimental models and different growing conditions and preparation protocols of algae sources, leading to variability in concentrations of molecules, degree of purity, or sample contaminations [37]. It has been proven that bacterial products usually contaminate crop-derived nutrients, thus resulting in nonspecific immune effects in vitro [15,50]. For these reasons, the experiments described in this study were performed with extracts obtained from *E. gracilis* cultured under in vitro standardized growing conditions, and extracts were prepared in pyrogen-free solutions, thus ruling out any possible nonspecific activities in cultured cells.

Systemic endotoxemia is characterized by leakage of LPS from the intestinal lumen into the bloodstream [22]. Usually, intestinal dysbiosis and inflammation are responsible for the reduction in sealing properties of the intestinal wall, but circulating levels of LPS are unexpectedly detected even in healthy conditions [51,52]. More significant, transient increase in circulating bacterial LPS triggers systemic inflammation characterized by macrophage recruitment, pro-inflammatory cytokine production, and alterations of hepatic, adrenal, and cardiovascular functions [52]. During bacterial endotoxemia, a plethora of soluble factors and highly reactive species are produced—each one is characterized by pathological effects that are not correctly counteracted by the anti-inflammatory and scavenger cellular mechanisms, amplifying the tissue damage. TNF-α is one of the first pro-inflammatory mediators released by macrophages following stimulation with LPS. Immediately after secretion, TNF-α orchestrates the pro-inflammatory cascade and via paracrine and autocrine signaling it activates various cell populations to amplify the inflammatory response. Therapeutic blockade of TNF-α has been proposed in different inflammatory-related conditions, including rheumatoid arthritis and inflammatory bowel disease [41,42]. However, prolonged microbial stimulation and chronic immune suppression exhaust tissue macrophages and reprogram cytokine production [53]. Dietary consumption of nutrients endowed with immunomodulatory activities should be a timely way to prevent the occurrence of intestinal dysbiosis and systemic inflammation while preserving macrophage function and survival of epithelial cells. In this study, long-term (4 days) exposure to *E. gracilis* extracts (i) did not affect macrophages responsiveness to LPS, (ii) maintained the anti-inflammatory effects of extracts (Table 3), and (iii) supplied the epithelial cells with reducing potential (Figure 5). Previous studies reported production by *Euglena* of α-tocopherol and glutathione able to change the redox status in cells. Moreover, culture conditions of microalgae such as carbon and nitrogen sources or tolerable stress induction can increase the yield of these biomolecules, further improving the nutritional importance of *E. gracilis* [37]. In morbid obesity, the increased supply of energy substrates and the inflammatory environment result in excessive mitochondrial ROS generation that finally contributes to the progression of obesity-related diseases such as atherosclerosis and type 2 diabetes. Conversely, low physiological levels of ROS, primarily generated at the plasma membrane and endomembrane, are required for normal cellular functioning and intracellular signaling [54]. Our study provides evidence that specific extract fractions of *E. gracilis* promote cellular reducing potential and avoid lipid peroxidation, conditions able to counteract LPS-induced oxidative burst (Figure 7). Moreover, our data reveal that different biological activities can be ascribed to various extracts of *E. gracilis* and that sub-fractionation can help target the anti-inflammatory and antioxidant effects. This study supports the role of *E. gracilis* supplementation in controlling intestinal bacteria-derived inflammatory conditions and the chronic disorders related to metabolic endotoxemia, but further studies are required for definitive conclusions.

## Figures and Tables

**Figure 1 microorganisms-09-02058-f001:**
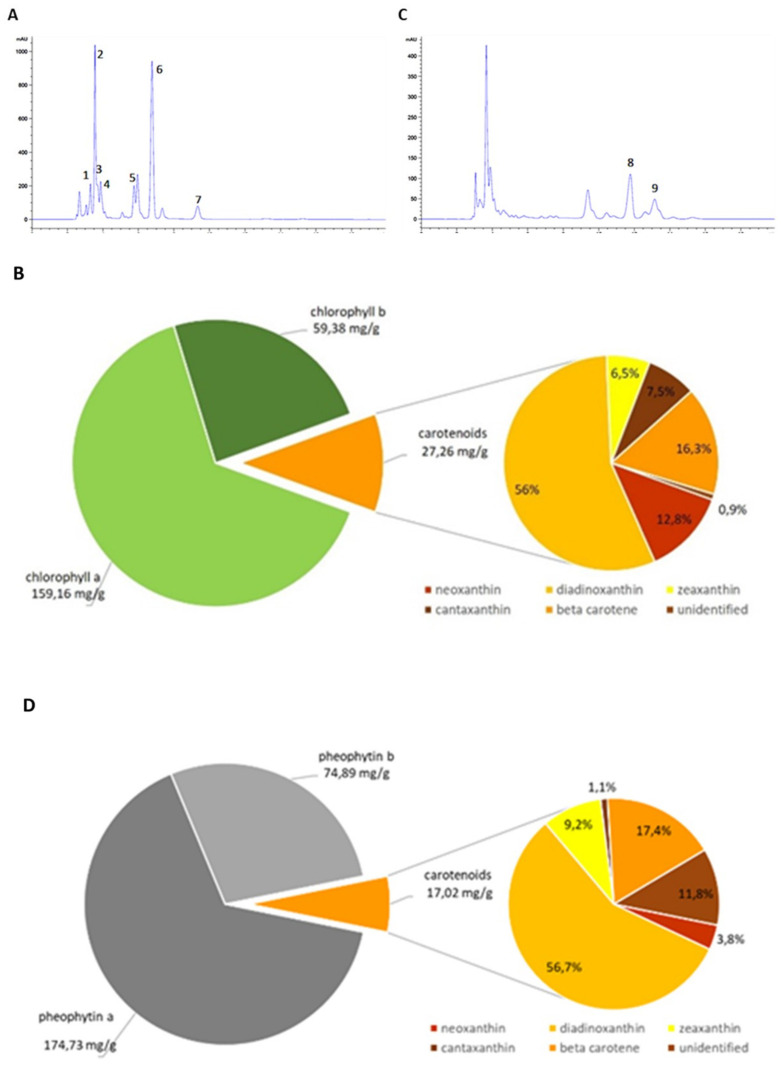
(**A**) Chromatogram of CE acquired at 438 nm; (**B**) chlorophyll a, chlorophyll b, total carotenoids expressed as mg/g of dry CE extract and relative carotenoid content. (**C**) Chromatogram of FE acquired at 438 nm; (**D**) pheophytin a, pheophytin b, total carotenoids expressed as mg/g of dry FE extract and relative carotenoid content.

**Figure 2 microorganisms-09-02058-f002:**
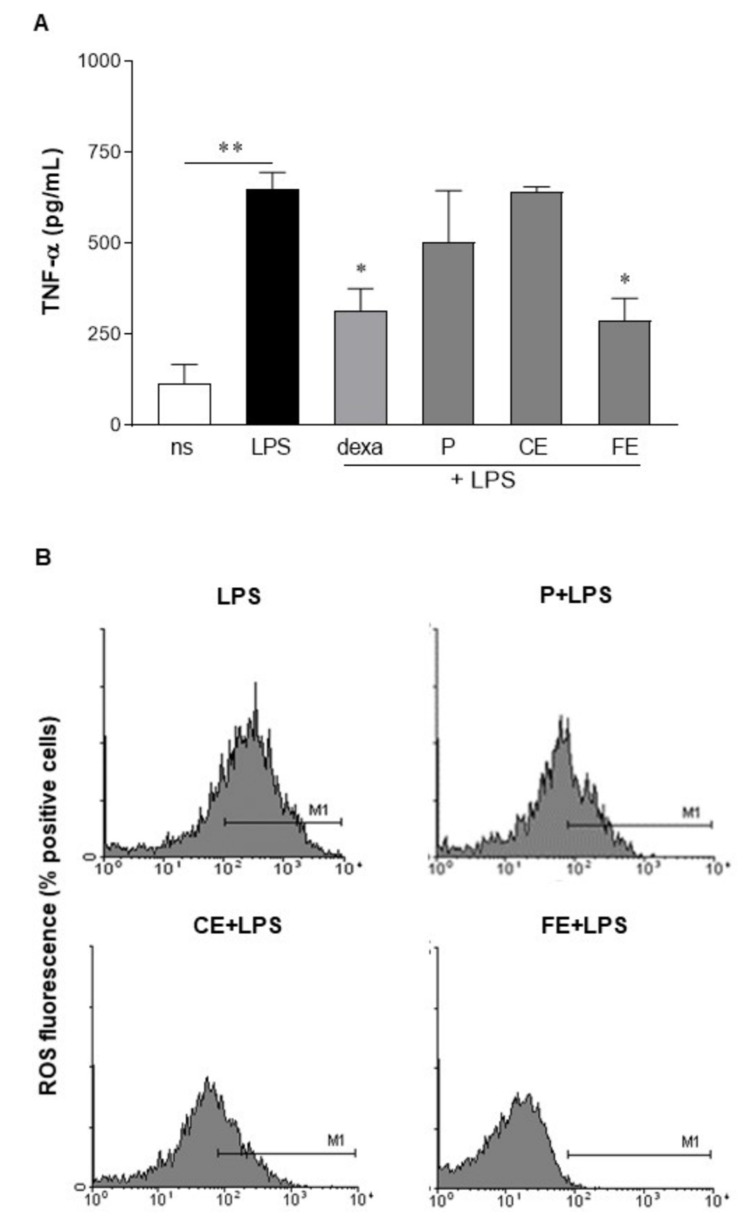
(**A**) Anti-inflammatory activity of extracts from *E. gracilis*. Primary human macrophages were stimulated for 24 h with 10 µg/mL of extracts obtained from *E. gracilis* and 100 ng/mL of lipopolysaccharide (LPS). TNF-α production was assessed in the conditioned media by ELISA. Data are reported as mean ± SE of the results collected in three independent experiments, each performed in triplicate. Dexa: dexamethasone 1 μg/mL. ** denotes *p* < 0.01 vs. non-stimulated cells; * denotes *p* < 0.05 vs. cells stimulated with LPS. (**B**) HT-29 cells were loaded with H2DCFDA and treated with extracts from *E. gracilis* and LPS (100 ng/mL). Positive cells were analyzed by cytofluorimeter. Representative histograms are reported.

**Figure 3 microorganisms-09-02058-f003:**
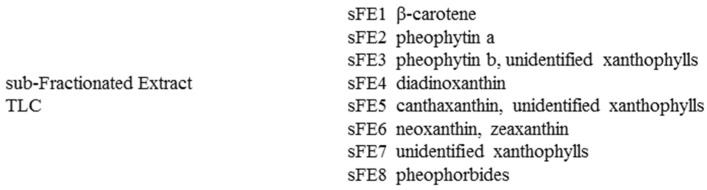
Schematic representation of the compounds identified in sub-fractionated extracts (sFE).

**Figure 4 microorganisms-09-02058-f004:**
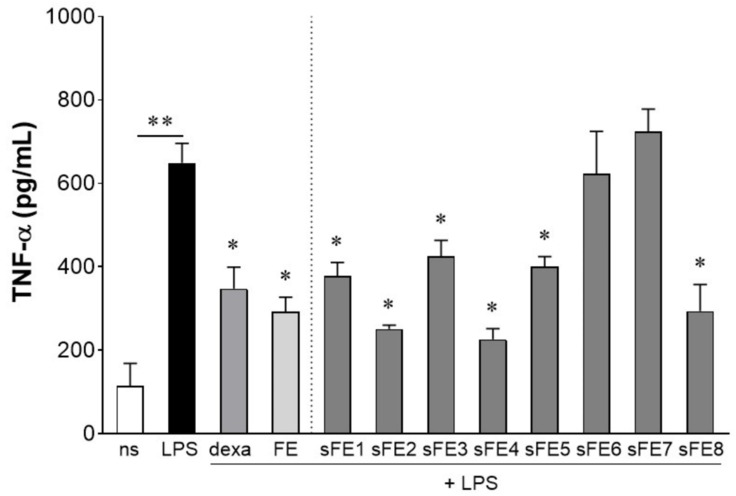
Anti-inflammatory activity of FE and sFE obtained from *E. gracilis*. Primary human macrophages were stimulated for 24 h with 10 µg/mL of extracts obtained from *E. gracilis*, 1 μg/mL dexamethasone (dexa), and 100 ng/mL of lipopolysaccharide (LPS). TNF-α production was assessed in the conditioned media by ELISA. Data are reported as mean ± SE of the results collected in three independent experiments, each performed in triplicate. ** denotes *p* < 0.01 vs. non-stimulated cells; * denotes *p* < 0.05 vs. cells stimulated with LPS.

**Figure 5 microorganisms-09-02058-f005:**
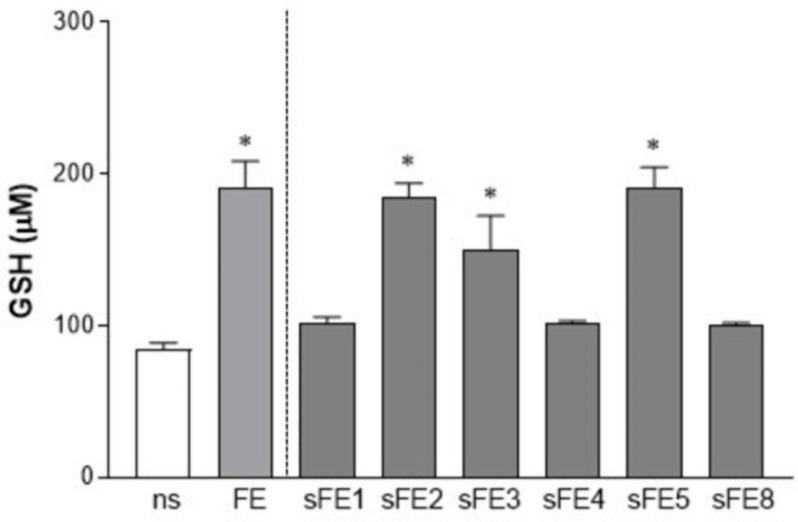
FE and sFE increase intracellular GSH levels. HT-29 were incubated with 10 µg/mL of extracts from *E. gracilis* for 4 days. Stimuli were renewed every day. Cells were then assessed for GSH content using Ellman’s method. Absorbance was recorded at 412 nm, and values were plotted on a standard curve obtained by serial dilution of *N*-acetylcysteine. Data are reported as mean ± SE of data obtained from two independent experiments, each performed in triplicate. * denotes *p* < 0.05 vs. non-stimulated cells (ns).

**Figure 6 microorganisms-09-02058-f006:**
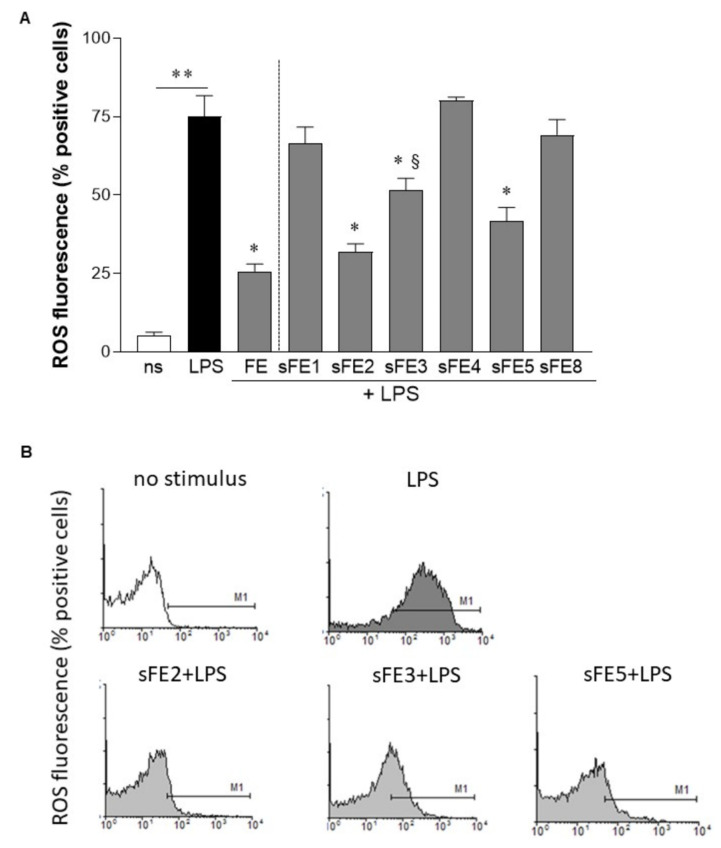
FE and sFE reduce intracellular reactive oxygen species generation. HT-29 cells were incubated with 10 µg/mL of FE or sFE for 3 days. Stimuli were renewed every day. Cells were then stimulated with LPS (100 ng/mL) for 24 h and then loaded with 10 µM 2′,7′-dichlorodihydrofluorescein diacetate (H_2_DCFDA), a fluorescent probe used to detect intracellular reactive oxygen species (ROS). Cells were analyzed by flow cytometry. For each experimental condition, 1 × 10^4^ events were collected. (**A**) Percentage of H_2_DCFDA-positive cells obtained from three independent experiments, each performed in duplicate. Data are reported as mean ± SE. ** denotes *p* < 0.05 vs. non-stimulated cells. * denotes *p* < 0.01 vs. cells stimulated with LPS. ^§^ denotes *p* < 0.05 vs. FE + LPS. (**B**) Representative histograms from cytofluorimetric analysis.

**Figure 7 microorganisms-09-02058-f007:**
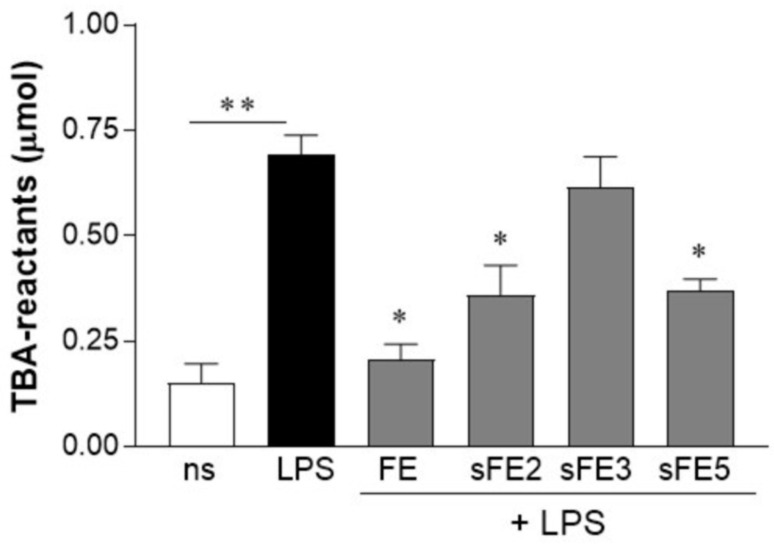
Pretreatment with FE and sFE prevents lipid peroxidation. HT-29 cells were incubated with 10 µg/mL of FE or sFE for 3 days. Stimuli were renewed every day. Cells were then stimulated with LPS (100 ng/mL) for 24 h. Thiobarbituric acid (TBA)-reactants were evaluated in cell lysates and normalized to protein contents. Data are reported as mean ± SE of results collected in three independent experiments, each performed in triplicate. ** denotes *p* < 0.01 vs. non-stimulated cells. * denotes *p* < 0.05 vs. cells stimulated with LPS.

**Table 1 microorganisms-09-02058-t001:** Identification of compounds in CE and FE fractions. Retention time (Rt), absorption maxima, and TLC Retention factor (Rf) of the identified compounds.

Compounds	Rt (min)	Absorption Maxima (nm)	Rf
neoxanthin (1)	3.3	435, 465	0.6
diadinoxanthin (2)	3.6	278, 445, 475	0.5
zeaxanthin (3)	3.7	275, 450, 476	0.6
canthaxanthin (4)	3.8	295, 475	0.65
chlorophyll b (5)	5.7	458, 617	0.7
chlorophyll a (6)	6.8	431, 663	0.68
β-carotene (7)	9.3	275, 451, 476	1
pheophytin b (8)	11.7	441, 650	0.75
pheophytin a (9)	13.1	410, 665	0.85
pheophorbides	3.6–3.8	441, 650; 410, 665	0.2

**Table 2 microorganisms-09-02058-t002:** Cell viability. Cell viability was determined in vitro using MTT test on differentiated primary human macrophages (Mφ) or HT-29 cells incubated for 24 h with extracts obtained from *E. gracilis*. IC_50_ is expressed in µg/mL.

Extracts from *E. gracilis*	Mφ IC50	HT-29 IC50
P	30	45
CE	48	57
FE	47	59
sFE1	>100	>100
sFE2	31	39
sFE3	33	41
sFE4	50	55
sFE5	47	58
sFE6	38	46
sFE7	37	42
sFE8	36	44

**Table 3 microorganisms-09-02058-t003:** Anti-inflammatory activity of extracts, dose–response assays. Human primary macrophages were stimulated with LPS 100 ng/mL and extracts (0–10 µg/mL) for 24 h. In different experiments, cells were incubated with extracts (0–10 µg/mL) for 3 days. Stimuli were renewed every day, and cells were finally challenged with LPS (100 ng/mL) for 24 h (total experimental time: 4 days). TNF-α was measured in the conditioned media by ELISA. Data are reported as the lowest concentrations (µg/mL) of *E. gracilis* extracts that significantly (*p* < 0.05) reduced by at least 25% the levels of TNF-α triggered by LPS stimulation.

Extracts *E. gracilis*	TNF-α 24 h	TNF-α 4 days
FE	1	2
sFE1	0.1	0.1
sFE2	1	1
sFE3	10	10
sFE4	2	2
sFE5	5	5
sFE8	1	2

## Data Availability

All data are available in the Manuscript.

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
