# Peer review of "Anti-Inflammatory Activities of Euglena gracilis Extracts"

_microorganisms, 2021, doi:10.3390/microorganisms9102058_

Round 1

Reviewer 1 Report

The authors report on the evaluation of anti-inflammatory properties of E.gracilis extracts and their chemical profile by HPLC analysis using UV-Vis detection. The most appropriate technique for the phytochemical investigation of this kind of extracts is the HPLC coupled online with both UV and ESIMS, which allows to establish the components present in mixture by selecting the total ion current chromatogram corresponding to the different mass value of each component. The reported analysis using only HPLC based on UV-vis detection can give only approximatively values on the quantitative presence of the components and it must be indicated in the text.

Considering the wavelength values ​​taken into account in the equations on page 3, the authors must specify why these are considered, while both in figure 1 and in figure S1. data are reported at 438 nm. They must standardize these measurements.

As expressly indicated in reference 29, the numerical values ​​of the equations reported at page 3 depend on the solvent and those chosen are for acetone 80% (aq v / v); a comment should be made on this aspect. For pheophytin a and b, verify that the values ​​indicated (653 nm and 665nm respectively) in the text are in agreement with the equations.

In table 1, Rf relating to TLC must be indicated in the caption; also specify that the numbers 1-8 for compounds refer to the numbering in Fig A and C. Replace commas with full stop in the numerical values.

Author Response

We would like to thank Reviewer 1 for the revision of our manuscript entitled “Anti-inflammatory activities of Euglena gracilis extracts”. We are now resubmitting the revised version of the Manuscript with all the revisions marked up using the Track Changes function. Moreover, we would like to respond in the following way to the points raised by the Reviewer 1:

“The most appropriate technique for the phytochemical investigation of this kind of extracts is the HPLC coupled online with both UV and ESIMS, … can give only approximatively values on the quantitative presence of the components and it must be indicated in the text.”

We agree with the Reviewer that the hyphenated HPLC-MS approach is nowadays the gold standard in quantitative analysis of phytochemicals and pharmaceuticals. In the present study, we dealt with a crude extract that, because of its complexity, results in matrix interferences caused by ion suppression and enhancement effects. These problems usually occur when matrix substances are co-eluted with the analytes of interest, thus compromising the accurate quantification of metabolites. Moreover, the instrument-related factors such as ion source parameters, transmission efficiency of ions, and detection efficiency make the hyphenated HPLC-MS a more “instrument dependent” approach compared to HPLC-DAD.

“Considering the wavelength values ​​taken into account in the equations on page 3, the authors must specify why these are considered, while both in figure 1 and in figure S1. data are reported at 438 nm. They must standardize these measurements.”

The Reviewer’s point is well taken. The equations reported on page 3 refer to chlorophylls and pheophytins that are characterized by absorbance maxima in the blue and red visible light. In particular, the absorbance maxima are observed at wavelength 663 nm for chlorophyll a, 646 nm for chlorophyll b, 665 nm for pheophytin a, and 653 nm for pheophytin b. The carotenoids are characterized by three absorbance maxima in the blue spectra, ranging from 400 to 500 nm. In the present study, for the quantification of carotenoids and for the respectively reported chromatograms, we chose the wavelength 438 nm as a fair compromise among the absorbance maxima of the carotenoids.

“As expressly indicated in reference 29, the numerical values ​​of the equations reported on page 3 depending on the solvent and those chosen are for acetone 80% (aq v / v); a comment should be made on this aspect. For pheophytin a and b, verify that the values ​​indicated (653 nm and 665nm respectively) in the text are in agreement with the equations.”

The Reviewer is right; the crude extract (CE) was obtained using acetone as a solvent for cell extract. Indeed, after evaporation to dryness, the pellet was dissolved in acetone 80% since preliminary experiments have shown that the dissolution of the pellet was more complete and fast as compared with other solvents. The water content (20%) was also useful in reducing the evaporation of the solvent during dilution and spectrophotometric analysis steps.

More comments about pheophytin a and b were added in the revised version of the Manuscript.

“In table 1, Rf relating to TLC must be indicated in the caption; also specify that the numbers 1-8 for compounds refer to the numbering in Fig A and C. Replace commas with full stop in the numerical values.”

As suggested by the Reviewer, we have revised paragraphs 2.2 and 3.1, accordingly.

Reviewer 2 Report

In this manuscript, Brun et al. describe ameliorative effects of Euglena gracilis extracts on LPS-induced ROS generation and inflammation in intestinal epithelial cell line (HT-29). The manuscript is well-written, and deeply discussed the related results, fairly simple set of experiments using typical and appropriate methodology. But, some minor mistakes should be revised, as follows, 1. On the contrary, we revealed that in cultured human primary macrophages and intestinal epithelial cell line, specific fractionated extracts of E. gracilis increased the cellular threshold for ROS activation under normal conditions leading to reduced oxidative burst and pro-inflammatory activation triggered by LPS. This meaning indicated that E. gracilis extract may be as a prooxidant. 2.Primary human monocytes were isolated from buffy-coat preparations of whole 152 blood obtained from the Hospital of Padova under a protocol approved by the University 153 of Padova Ethics Committee. Please added the Institutional Review Board Statement 3.How about the DMSO concentration in each cell study? 4. Please add the calculation equation of ROS. 5. Please clarify the p < 0.02, why not p < 0.01 6. R2 chaged to R2 (R squared)

Author Response

We would like to thank Reviewer 2 for the revision of our manuscript entitled “Anti-inflammatory activities of Euglena gracilis extracts”. We are now resubmitting the revised version of the Manuscript with all the revisions marked up using the Track Changes function. Also, we would like to respond in the following way to the points raised by the Reviewer 2:

1. On the contrary, we revealed that in cultured human primary macrophages and intestinal epithelial cell line, specific fractionated extracts of E. gracilis increased the cellular threshold for ROS activation under normal conditions leading to reduced oxidative burst and pro-inflammatory activation triggered by LPS. This meaning indicated that E. gracilis extract may be as a prooxidant.”

We thank the Reviewer for this observation. The role of the reactive species in cell metabolism and response is still controversial especially when cells (immune or epithelial cells) undergo environmental stressors such as bacterial infections (in our experiments, we used LPS to mimic the exposure to bacterial products). The reactive species of oxygen (ROS) have an active role in most of the signaling cascades involved in cell development, proliferation, and survival, acting as important second messengers that are of critical importance during diseases. In the gut, cells are usually under the assault of luminal bacterial products thus providing a low but continuous grade of stressors that have been reported to be beneficial for the host health. This hormetic effect of ROS increases the threshold for cell activation. At the same, in this study, we demonstrated that E. gracilis increases the cellular threshold for ROS activation under LPS-free conditions thus making cells ready for possible microbiological insults.

However, we disagree with the Reviewer, as we cannot consider E. gracilis as pro-oxidant. Indeed, in several tissues and cells, the oxidative damage correlates with protein and lipid peroxidation. Actually, we observed lipid peroxidation only in LPS-challenged cells, not in cells stimulated with extracts from E. gracilis alone.           

“2.Primary human monocytes were isolated from buffy-coat preparations of whole blood obtained from the Hospital of Padova under a protocol approved by the University of Padova Ethics Committee. Please added the Institutional Review Board Statement.”

The statement was added in the revised version of the Manuscript.

 “3.How about the DMSO concentration in each cell study?”

In each sample, the DMSO concentration was below 0.01% vol/vol. As reported in the Manuscript, control cells were stimulated with the same vehicles as extracts at the highest final volumes used and, therefore, the cells were exposed to the same concentrations of DMSO.

“4. Please add the calculation equation of ROS.”

No equation was used to calculate ROS. Data are reported as the percentage of the cells positive to the intracellular H2DCFDA fluorescent probe. The percentage is directly obtained during the FACS analysis.

“5. Please clarify the p < 0.02, why not p < 0.01”

The Reviewer is right. 0.02 was corrected in 0.01 in the revised version of the Manuscript.

“6. R2 chaged to R2 (R squared)”

Done.

Thank you in advance for your attention.

Round 2

Reviewer 1 Report

  The reason reported in the reviewer's answer on the choice of the value of 438 nm must be added in the text.
Afetr that, I recommend to accept the revised version .

This manuscript is a resubmission of an earlier submission. The following is a list of the peer review reports and author responses from that submission.

Round 1

Reviewer 1 Report

The manuscript entitled „Anti-inflammatory activities of Euglena gracilis extracts” presents the results of some biological experiments made on Euglena gracilis samples. The article is basically well written, logically structured, and correspondingly presented.

Questions to be addressed:

  • What was the selection criteria for aceton used for extraction? Did the authors perform preliminary investigations to determine the best solvent for the extraction of Euglena gracilis samples? How about other organic solvents possibly suitable for these experiments?
  • Depending on the solvents used the chemical profile of the extracts can greatly vary, which significantly influences the biological activity
  • The authors should provide HPLC and /or TLC chromatograms of the identification process
  • I miss the quantification of the identified secondary plant metabolites to see which are the major compounds.
  • It would be great to determine the contribution of each major constituent of Euglena gracilis extract to the overall pharmacological activity

Reviewer 2 Report

The authors report a study on the anti-inflammatory activities of extracts obtained from a cultured sample of E. gracilis. The biological evaluation is rigorous, but applied to fractions of the extract whose chemical composition has been very approximated.

The chemical profile and extract fractioning is very and very poor and misleading. At the present, TLC and a simple HPLC qualitative analysis cannot be sufficient to characterized the presence of metabolites. It is true that the study of the metabolites of E. gracilis has been little studied, but the authors do not cite two relevant works: Fumio Matsuda et al., Biosci Biotechnol Biochem 2011 and Lory Z Santiago-Vázquez et al., J Chromatogr B Analyt Technol Biomed Life Sci, 2004. In particular, in the latter paper hydroxyl fatty acids were identifies by HPLC-MS analysis, and this is the most appropriate technique to study the chemical profile.

Figure 1 is redundant and not very explanatory; the description must be reported in a proper paragraph, indicating the amounts of crude extract and of the fractions obtained. Furthermore, a table must be inserted indicating the retention time, UV lambda values and the bibliographic references used to identify the constituents of each fraction. In Supplementary, the chromatograms of each fraction must be given.

Anyway, the applied analytical technique remains not fully suitable for the identification of the metabolites; e.g. in SFE1 beta-carotene is indicated as the only component, whereas in the discussion the authors report on beta- carotenoids.

Paramylon isolation has been carried out, but no analytical details are given for characterizing it. The authors report: “Moreover, acetone precipitates paramylon [31], 373 giving us the possibility to test the antioxidant and anti-inflammatory activities of E. gracilis in paramylon-free fractions. Whereas paramylon has been indicated as the most important antioxidant fraction in E. gracilis, in this study we were unable to associate any anti-inflammatory and antioxidant activities to the paramylon fraction”, without finding an explanation or verifying the effective presence of paramylon in the tested fraction.

In the biological analyses, a known reference compound must be added as control test and introduced in the figures.

In summary, the work cannot be accepted in this form, which besides being lacking is misleading and it needs a strong implementation to be accepted for publication.v

The report from the file attached by this reviewer

"The authors reported an overview on metabolites from soft coral genus Sarcophyton and its associated marine fungi. They should point out the novelty and relevance of their work, especially in comparison with 
the 15 review articles published previously, from 1989 to 2020, as deduced by a bibliographic research on Chem Abstr. online.The period covered by this overview should also be specified in the Introduction. Regarding the chemical approach of the isolated metabolites, details are reported on the NMR technique used for the structural characterization, but also the conditions of their isolation (eg. solvent extractions, HPLC purification) must be included. Molecular structures give indication of stereocenters, but assignments of relative and mainly absolute configurations are of interest and must be included and discussed.

Abbreviated titles of references have to be used e.g. for references 6,10,13, 14 and a number of following ones"